# Effects of Nursing Simulation Using Mixed Reality: A Scoping Review

**DOI:** 10.3390/healthcare9080947

**Published:** 2021-07-27

**Authors:** Kyeng-Jin Kim, Moon-Ji Choi, Kyu-Jin Kim

**Affiliations:** 1Department of Nursing, Kyungil University, Gyeongsan 38428, Korea; kkj0908@kiu.kr (K.-J.K.); cmoonj12@kiu.kr (M.-J.C.); 2Daegu Center for Infectious Diseases Control and Prevention, Daegu 41940, Korea

**Keywords:** mixed reality, simulation, nursing

## Abstract

Mixed reality (MR) has recently been suggested as a new educational tool for nursing simulation. However, few studies have assessed the use and evaluation of MR nursing simulations. Therefore, this review identified studies of MR nursing simulations through a scoping review using the framework proposed by Arksey and O’Malley. The studies reviewed were found through DBpia, RISS, PubMed, CINAHL, and Google Scholar. Each study was analyzed, and data were abstracted into publication characteristics, simulation program details, device use, and simulation evaluation. A total of 10 studies were reviewed. Senses that were mainly used in MR nursing simulation included audition and haptics via motion, in addition to visual information. Simulations were evaluated using various outcome variables. Knowledge was most commonly evaluated, followed by clinical performance and satisfaction. This study is significant as it analyzed trends in research on MR nursing simulations in Korea and other countries and suggested directions for the use of MR technology in nursing simulations based on the findings. Additional studies are required to develop scenarios for the effective use of MR in nursing education and to evaluate the effects of MR nursing simulations.

## 1. Introduction

The coronavirus disease (COVID-19) pandemic has led to changes in the paradigm of medicine as well as politics, economy, society, and culture. The need for online education in nursing has increased with recent interest in limiting contact between persons, and the increased demand for alternative environments in clinical practice indicates the importance of changes in education in the post-COVID-19 era [1]. This emphasizes the demand, importance, and necessity of simulation education as a complementary method to increase the experience of independent direct nursing in complex clinical settings [2,3].

Simulation education creates situations similar to those encountered in reality, but in a safe learning environment, and learners use their knowledge through their own judgments [3]. Nursing simulation is an education method that enables learners to improve their knowledge of nursing, critical thinking, and problem-solving skills through debriefing the process after solving problems in different simulated scenarios that may occur in clinical settings [2]. According to the theoretical framework of Jeffries [4], education, intervention, and evaluation of a simulation are achieved through the interaction between the instructor and learner, and the simulation can be evaluated by assessing knowledge, performance, satisfaction, critical thinking, and confidence. However, most previous simulation practices in nursing have been conducted in school practice rooms. Simulation education in such settings requires a separate space and management to install simulators or additional training for standardized patients. As a result, additional time and money are required, and the same sections need to be repeated several times because only a small number of students can participate in each session [5]. Therefore, simulation using virtual reality (VR) or mixed reality (MR) is an alternative to overcome these shortcomings. MR simulation is a method of converging the virtual and real worlds using devices such as computers, and it can provide education regardless of the location. Users can view the learned activities from different aspects, and MR increases the engagement of users in education through the simulation of various scenarios. MR simulation education is not costly, unlike high-fidelity simulation, and is flexible regarding environmental constraints, as it can be conducted in any space with a computer. In addition, the user can repeat the learning individually, reducing the time constraints for the instructors.

According to the reality-virtuality continuum, MR is defined as a concept that encompasses both the real world and the virtual environment [6]. However, focus has typically been on visual displays, while the real world consists of more than visual stimuli. Thus, the definition of mixed reality has often been limited. Speicher et al. [6] identified five factors in addition to visual stimulation: audition, motion, haptics, taste, and smell. In addition, experts have defined MR as a combination of reality and virtuality and as strong augmented reality (AR), rather than as AR that uses specific hardware, such as HoloLens. Therefore, MR can reproduce CG images at an interactive level, as if they were part of the real world. The concept of MR is different from that of AR, which provides additional information to reality using only visual information. MR is also distinct from immersive VR, which completely blocks the external world [7]. MR is understood as a technology that is combined with AR; this immersive technology was selected as one of the top 10 technology trends in 2018, with the expectation that it would contribute greatly to relevant fields. Previous studies have reported anatomy education using AR with only visual images and no interaction [8] and VR simulation using computers with a head mount display (HMD) [3,9]. However, few studies have used MR for nursing simulation education, and MR is primarily used for skill training of doctors, when conducting learning procedures such as airway management training and surgeries [10,11].

The faculty of nursing in Korea provides a curriculum aimed at developing step-by-step knowledge and skills, which is divided into clinical nursing and basic nursing, including basic science [12]. Basic science teaches a large amount of anatomy and microbiology and has a high utilization of VR and AR, which has the advantage of increasing the understanding and satisfaction of nursing students [13]. In addition, pre-clinical training courses are suitable for many students to simultaneously access online classes [14]. However, they are not suitable for practicing skills directly, making decisions, and dealing with certain clinical situations. Nurses should be trained to make accurate clinical decisions through critical thinking include the performance of knowledge or skills [12]. During a global health crisis like the COVID-19 pandemic, virtual or mixed platforms can be useful for repeated training or preparation of nurses. In particular, 73% members of the National Nurses Association (NNA) reported that they had discontinued their undergraduate nursing education as a result of the social distancing measures implemented due to COVID-19, and 46% of nursing students experienced clinical placement cancellations during the same period [15,16]. The COVID-19 pandemic has disrupted and transformed nursing education systems in many countries, requiring the creation of different nursing platforms through the design of efficient educational programs.

Therefore, analyzing the research trends and examining the characteristics of MR simulation programs would be helpful to find a strategy for MR nursing simulation programs and to prepare plans to effectively present the results of programs in the future. The purpose of this study was to analyze research trends in Korea and worldwide from 1990 to April 2021 in MR simulation program education, using a topic range review method to provide basic data for the use of MR simulation programs in nursing education. This study investigated research trends related to MR simulation by analyzing the characteristics and effects of the applied education program and identified effective methods and strategies in addition to the outcome of education.

## 2. Materials and Methods

### 2.1. Research Design

This was a scoping review that assessed trends in Korean and overseas studies of MR nursing simulation.

### 2.2. Research Participants and Data Collection

The goal of a scoping review study is to investigate the scope and essence of a research topic and summarize the research outcomes or gaps in existing studies. This study was conducted according to the five research stages of scoping review, as suggested by Arksey and O’Malley (2005) [17].

#### 2.2.1. Stage 1: Identifying the Research Question

The population, concept, and context of this study were established based on the guidelines for systematic scoping reviews by Peters et al. [18] to analyze the current status, methods, and effects of nursing simulation education using MR. The population consisted of nursing students and nurses. The concept was a simulation program using MR, and the context was nursing. The research question was, “What are the trends in Korean and international research studies of MR nursing simulation, and what directions could help nursing education?” The goal was to assess the methods of simulation programs, as well as methods used to assess simulation effects, and to conduct a program evaluation.

#### 2.2.2. Stage 2: Identifying Relevant Studies

Studies published in Korea and other countries from 1990 to April 2021 were sought. In 1994, Paul Milgram referred to everything between virtuality and reality as mixed reality (of which augmented reality represents a class on the reality-virtuality continuum), establishing a specific definition of mixed reality [6]. Therefore, only studies published between 1990 and 2021 were included in this study. A literature search was conducted from 10 May to 26 May 2021. For studies published in Korea, “mixed reality,” “nursing,” “simulation,” and “HoloLens” were set as search keywords, and studies were sought via the Research Information Sharing Service (RISS), DataBase Periodical Information Academic (DBpia), and Korean Studies Information Service System (KISS). Studies published outside of Korea were sought in PubMed, Cumulative Indexing Nursing and Allied Health Literature (CINAHL), and Google Scholar. The search keywords included a combination of “Mixed Reality,” “HoloLens,” “Magic Leaf,” “Nursing,” and “Simulation” (Appendix A).

The selection criteria for studies were those published in academic journals from 1990 to April 2021. The criteria for data selection were studies published in Korean or English related to immersive MR with nurses or nursing students. Conference posters and abstracts, books, and studies that did not specify the study participants, and studies that did not use MR tools and methods were excluded. In addition, research papers based on the same scenario and content by the same author were excluded (Table 1).

#### 2.2.3. Stage 3: Study Selection

A total of 2868 studies were obtained, including 4 on DBpia, 9 on RISS, 0 on KISS, 242 on PubMed, 48 on CINAHL, and 2565 on Google Scholar. A total of 1794 studies that were duplicates were excluded. Three independent researchers reviewed the remaining 1074 articles. The title and abstract were reviewed first to select 443 papers, excluding 631 studies whose purpose, participants, and content did not meet the selection criteria. A total of 418 studies were selected, excluding 25 studies whose full text was not available. Studies for final review were selected by confirming the selection and exclusion criteria after individual reviews and meetings of the researchers. A total of 408 studies were included in this analysis. The data collection and selection processes are shown in Figure 1.

### 2.3. Data Analysis

#### 2.3.1. Development of Framework for Systematic Analysis

For systematic analysis of studies of education for nurses and nursing students, the analysis criteria used in literature related to new nurse education programs were reviewed; accordingly, an analysis framework was established. The framework consisted of publication- and research-related educational programs and program evaluation characteristics to analyze the characteristics of the studies. First, publication- and research-related characteristics included four sub-categories: year of publication, study design, study population, and sample size. The study design was divided into quantitative, qualitative, mixed, and methodological studies. Second, the MR nursing simulation characteristics consisted of the development platform, interaction device, intervention scenario, and intervention time. Third, evaluation of the simulation program was categorized into knowledge, performance, satisfaction, critical thinking, and confidence according to the theoretical framework of Jeffries [4]. The other variables used in the study were grouped into sub-categories to establish the development framework.

#### 2.3.2. Quality Evaluation of Selected Studies

To maintain the methodological rigor of the study, a quality assessment tool suggested by Hawker et al. [19] was used to evaluate the quality of the selected studies. The three researchers read all the studies, discussed inconsistencies, and agreed with the interpretation of the studies before proceeding to the next step in order to ensure reliability and consistency of the overall analysis process. Quality evaluation was according to the categories good, fair, poor, and very poor, as described in Birtill et al. [20] (Appendix B).

### 2.4. Stage 4: Data Categorization

Data were entered into a development framework consisting of three categories of publication and research-related characteristics, education program characteristics, and program evaluation characteristics, and 16 sub-categories using Microsoft Excel 2016. The collected data were encoded and analyzed using descriptive statistics (frequencies and percentages) in Excel.

### 2.5. Quality Evaluation of the Current Study

To ensure a systematic and consistent scoping review, quality evaluation was conducted using the Preferred Reporting Items for Systematic reviews and Meta-Analyses extension for Scoping Reviews (PRISMA-ScR) checklist [21].

## 3. Results

### 3.1. Publication- and Research-Related Characteristics of the Literature

A total of 10 studies were used in the final analysis; the results are shown in Table 2. Three, two, and two of the analyzed studies were published in 2018, 2017, and 2020, respectively. Five of the studies were quantitative. Two studies were qualitative, and another two were methodological studies of program development. The number of participants in the studies varied from 5 to 107, depending on the study design. The sample size was not mentioned in one methodological study. Six (60.0%) and four (40.0%) studies were conducted with nursing students and nurses, respectively.

### 3.2. Mixed Reality Nursing Simulation Characteristics

The Unity engine was most frequently used as the development platform. Microsoft HoloLens was most frequently used as the visual device. Other sensory information included auditory displays and haptics. The average simulation time ranged from 5 min to 300 min, and YouTube videos were used for pre-training before the simulations. Scenarios used in the MR nursing simulation included five, three, and two simulations of judgments in fire or emergency situations, patient assessment, and procedures and treatment, respectively (Table 3).

### 3.3. Evaluation of Mixed Reality Simulation

Table 3 shows the results of the analysis of 10 simulation programs used in the selected studies, according to Jeffries’ evaluation method [4]. Knowledge was assessed in four studies. Clinical performance, satisfaction, critical thinking, confidence, and other variables were evaluated in three, three, zero, two, and five studies, respectively. Other evaluation variables included assessment of patient safety, assessment of needs, motivation, and team activity, as well as side effects such as motion sickness.

## 4. Discussion

This scoping review investigated the current trends and research status through a literature review of 10 selected studies of mixed reality simulation education published in Korea and other countries and assessed the simulation methods, devices, and evaluations.

MR nursing simulation has primarily been assessed in 10 studies in the last five years since 2015. This may be attributed to recent developments in science and technology, leading to the commercialization of devices for MR technology and platforms that allow for interactions with virtual environments. In fact, virtual training systems generally use VR technology; the lack of devices with appropriate viewing angles or usage time in AR or MR has led to the limited use of MR nursing simulation [32]. Studies initially assessed AR, focusing only on the visual sense; however, the introduction of the HoloLens by Microsoft in 2016 led to extensive subsequent development [7].

The device most frequently used for MR nursing simulation was the HoloLens, and visual and auditory information was generally provided through that device. In addition, tactile stimulation was provided using a mannequin or haptic device to enhance the effects of the simulation program. Nursing is a human-centered care process, and dynamic feedback is observed between patients and nurses [33]. Therefore, it is essential to provide auditory, haptic, and olfactory senses in addition to visual senses to increase engagement in simulation scenarios, including training to assess the main complaints of the patients. It is necessary to further develop and expand MR nursing simulations to provide integrated sensory information.

Scenarios that were predominately used in mixed reality nursing simulation included five, three, and two simulations of assessments of fire or emergency situations, patient assessments, and procedures and treatment, respectively. This finding suggests that MR can be used in clinical nursing as technology is being developed to integrate virtual and real spaces. In addition, developing and materializing various scenarios into MR would be helpful for educational interventions that could expand the experience of nursing students and nurses in performing problem-centered nursing processes for different patients, thereby strengthening their competency as nurses.

The average nursing simulation time using the MR varied from 5 to 300 min. MR can compensate for the shortcomings of VR, including motion sickness and dizziness caused by being isolated from the external world. Therefore, MR can be used for a longer period of time compared to VR and would be highly useful for continuous nursing processes from inspection to evaluation of patients. In addition, MR technology is integrated with multisensory channels and surrounding environment sensors. Thus, it would be appropriate for nursing education that requires a combination of clinical judgment and skills [7].

A previous study reported that considerable time is required to prepare in advance before use of immersive devices such as VR and MR [34]. However, preparation was only observed in two studies. Furthermore, most pre-education was conducted using YouTube videos. In most simulation education, preparation is fundamental [35]. In particular, when using a new device, preparation is more important because limited understanding of the device can interfere with the user’s engagement in the simulation. Therefore, pre-education is critical for MR simulation, and it is necessary to help learners to understand MR simulations in advance.

The effects of the MR nursing simulation were evaluated by measuring knowledge, clinical performance, satisfaction, and confidence. Other variables included motivation, side effects (e.g., motion sickness), cost effectiveness, and demand. In a meta-analysis of the effects of immersive VR simulation programs [3], the outcome variables of VR simulation were generally knowledge and clinical performance, while critical thinking or confidence were not evaluated. Simulation programs have not been widely evaluated, as studies of future education using VR and MR have not actively been conducted. In the future, studies are needed to evaluate critical thinking and confidence. MR reproduces situations similar to real environments by applying virtual CG to virtual or real environments and allows users to feel that they are experiencing scenarios as in reality. This improves engagement and satisfaction and increases critical thinking abilities through judgments of the given situation. In virtual simulations, haptic information is not available. Thus, MR compensates for the shortcomings of VR and mixes the virtual environment with real senses to improve clinical performance and critical thinking.

In this study, research trends in MR simulation education in Korea and elsewhere were reviewed using a scoping review method. Based on these findings, new directions for nursing stimulation were suggested. However, mixed reality nursing simulation was analyzed using studies published in academic journals; thus, there is a limitation in reflecting the actual conditions of stimulation programs that are currently used. With increasing interest in contact-free interactions, it is necessary to provide policy support for MR nursing simulation programs for future education. Moreover, it is important to develop and apply various simulation scenarios and develop systematic simulation programs to provide evidence-based education according to the theoretical framework for simulation evaluation.

## 5. Conclusions

Following the increasing interest in contact-free interactions, interest in VR/AR/MR has gradually escalated. MR nursing simulation scenarios generally included situations that required judgment, but there were few scenarios that addressed procedures and techniques. The evaluation of simulation programs was generally conducted by measuring knowledge, and there was insufficient assessment of critical thinking.

Based on the findings of this study regarding trends and characteristics of studies of MR simulation education, the following suggestions are made. Nursing education simulation requires programs that integrate both nursing skills and situational judgment, which are essential traits for nurses. MR can properly utilize and reproduce virtuality and reality. Thus, MR may be effective in nursing education simulations. In addition to the development of various educational programs, future studies would need to establish evaluation systems and develop programs by stage to assess the effects of simulations.

## Figures and Tables

**Figure 1 healthcare-09-00947-f001:**
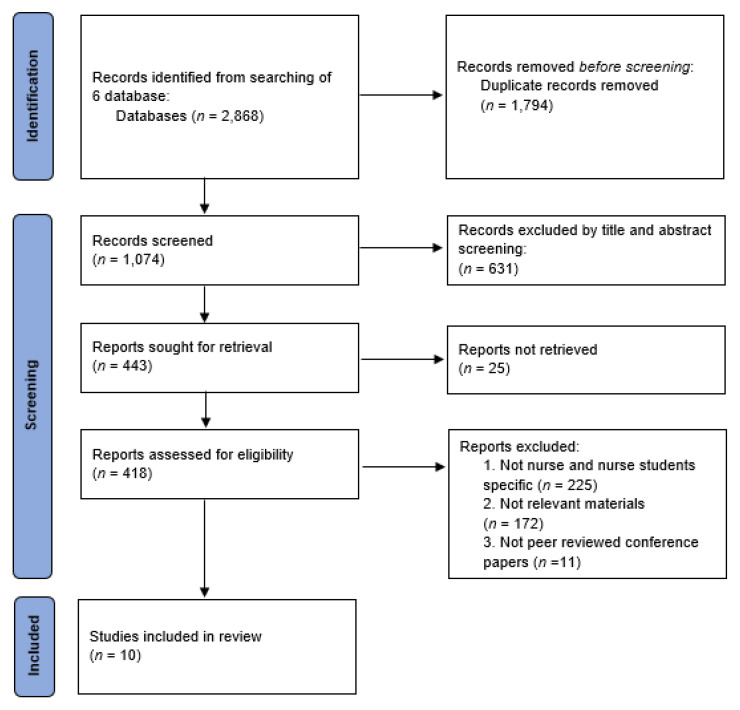
Study selection flow chart.

**Table 1 healthcare-09-00947-t001:** Inclusion and exclusion criteria.

Criteria	Specified Criteria
Inclusion	▪ Studies published from 1990 ▪ English and/or Korean studies ▪ Studies of nurses or nursing students ▪ Nursing simulation
Exclusion	▪ Abstract ▪ Review including systematic review, meta-analysis, etc. ▪ Non-English and/or Non-Korean studies ▪ Non peer-reviewed articles ▪ Not nursing simulation ▪ Unclear description of MR tools and methods ▪ Virtual reality study with only visual information

**Table 2 healthcare-09-00947-t002:** Publication characteristics and research design of included studies (*N* = 10).

Variables	Categories	*N* (%)
Year of publication	2015	1 (10.0)
	2016	1 (10.0)
	2017	2 (20.0)
	2018	3 (30.0)
	2019	1 (10.0)
	2020	2 (20.0)
Research design	Quantitative	5 (50.0)
	Qualitative	2 (20.0)
	Mixed method	1 (10.0)
	Methodological	2 (20.0)

**Table 3 healthcare-09-00947-t003:** Summary of characteristics of studies included in scoping review (*N* = 10).

	Author	Year	Study Objective	Methods (a) Sample (*N*) and (b) Study Design	Immersive-Mixed Reality Simulation							
					Device Used (a) Development Platform (b) INTERACTION Device	Intervention (a) Scenario (b) Time (Min)	Measurement					
							M1	M2	M3	M4	M5	M6
1	Alonso et al. [22]	2017	To assess the effect of telematics support in CPR situation	(a) 72 nurses (cont. = 36, int. = 36) (b) Quantitative study	(a) wHealth Live Streaming (b) V (Google Glass), A (USB headset Mono Earbud), H (SimMan 3G)	(a) CPR coaching		●				
2	Byrne and Senk [23]	2017	Hands free SBAR communication	(a) 11 baccalaureate nurses (b) Mixed methods pilot study	(b) V, A (Google Glass)	(a) Blood transfusion (b) Prepared 3 min YouTube						●
3	Frost et al. [24]	2020	To assess holo-patient’s pain (holographic image)	(a) 13 health care students (b) Qualitative study pilot study	(a) Gigxr (b) V, A (Microsoft HoloLens)	(a) Myocardial infarction (b) 15 min	●					●
4	Hauze. et al. [25]	2018	To compare a written case study and holographic mixed reality simulation	(a) 107 baccalaureate nursing students (cont. = 54, int. = 53) (b) Quantitative study	(a) Unity engine (b) V, A (Microsoft HoloLens)	(a) Anaphylaxis	●		●		●	●
5	Lai & Chang [26]	2018	To test and evaluate a prototype of mixed reality simulation	(a) 10 nursing students (b) Methodological study pilot study	(a) Unity engine (b) V (Google Cardboard), M (Leap motion sensor)	(a) Nasogastric tube care	-	-	-	-	-	-
6	Teng et al. [27]	2019	Multiple patients in ICU, real-time monitor V/S, and other conditions	(a) Nursing student (b) Methodological study	(a) Unity 3D & Magic Leap Lumin SDK package (b) V, A (Microsoft HoloLens), M (Magic Leap One)	(a) ICU patient monitoring	-	-	-	-	-	-
7	Vaughn et al. [28]	2016	To determine the effect of high-fidelity simulation on students.	(a) 12 prelicensure students (b) Quantitative study pilot study	(b) V (Google Glass), H (Manekin)	(a) Lab scenario (b) 10 min pre-brief (video)	●	●	●		●	
8	Wuller et al. [29]	2018	To test effect of Smart Glasses applications on nurses’ work.	(a) 5 nurses (b) Qualitative study	(b) V, A (Vuzix M100)	(a) Wound care						
9	Wunder et al. [30]	2020	To evaluate the skills of SRNAs who participated in fire simulation	(a) 32 registered nurse anesthetists (b) Quantitative study, descriptive study	(b) V, A (Magic Leap One)	(a) Fire in the operating room (b) 5~14 min/scenario		●				●
10	Yoshida et al. [31]	2015	Effect of intraoperative use of the HMD on scrub nurses.	(a) 15 scrub nurses (b) Quantitative study, prospective study	(b) V, A (AiRScouter)	(a) Novel intraoperative instructional tool (b) 300 min (147–400 min range)	●		●			●

A = auditory; H = haptic; M = motion; V = visual; CPR = cardiopulmonary resuscitation; HMD = head-mounted display; ICU = intensive care unit. Outcomes: M1 = knowledge, M2 = skill performance, M3 = learner satisfaction, M4 = critical thinking, M5 = self-confidence, M6 = other.

## Data Availability

Not applicable.

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
