# Peer review of "Effects of Nursing Simulation Using Mixed Reality: A Scoping Review"

_healthcare, 2021, doi:10.3390/healthcare9080947_

Round 1

Reviewer 1 Report

This is a very topical scoping review which is conducted according to a standard 5-step method for scoping reviews and also adheres to PRISMA principles. The results are very clearly presented in the table and this will be a useful study for further research and for others practicing and working in this area. I only have a minor suggestion which is, in the abstract, do not say 'databases such as...'; state them here precisely.

Reviewer 2 Report

The manuscript entitled Effects of Nursing Simulation using Mixed Reality: A Scoping Review report an interesting scoping review on th educational approacches to nursing training. 

Nurses with the COVID-19 pandemic have become one of the the first lines of soldiers in the health systems and therefore it is clear how important is their training and preparation. 

I would therefore contextualize first the nursing curricula and how difficult is for these students the approach to the first-year disciplines 

https://www.sciencedirect.com/science/article/pii/S1557308721000573

https://www.mdpi.com/2076-3417/10/7/2357

and then how covid 19 made this training even more difficult 

https://mededu.jmir.org/2020/2/e20963/?utm_source=TrendMD&utm_medium=cpc&utm_campaign=JMIR_TrendMD_1
